# Affordance-Based Grasping Point Detection Using Graph Convolutional Networks for Industrial Bin-Picking Applications

**DOI:** 10.3390/s21030816

**Published:** 2021-01-26

**Authors:** Ander Iriondo, Elena Lazkano, Ander Ansuategi

**Affiliations:** 1Department of Autonomous and Intelligent Systems, Fundación Tekniker, Iñaki Goenaga, 5-20600 Eibar, Spain; ander.ansuategi@tekniker.es; 2Computer Science and Artificial Intelligence (UPV/EHU), Pº Manuel Lardizabal, 1-20018 Donostia-San Sebastián, Spain; e.lazkano@ehu.eus

**Keywords:** affordance grasping, grasping point detection, graph convolutional network, pick and place, deep learning

## Abstract

Grasping point detection has traditionally been a core robotic and computer vision problem. In recent years, deep learning based methods have been widely used to predict grasping points, and have shown strong generalization capabilities under uncertainty. Particularly, approaches that aim at predicting object affordances without relying on the object identity, have obtained promising results in random bin-picking applications. However, most of them rely on RGB/RGB-D images, and it is not clear up to what extent 3D spatial information is used. Graph Convolutional Networks (GCNs) have been successfully used for object classification and scene segmentation in point clouds, and also to predict grasping points in simple laboratory experimentation. In the present proposal, we adapted the Deep Graph Convolutional Network model with the intuition that learning from *n*-dimensional point clouds would lead to a performance boost to predict object affordances. To the best of our knowledge, this is the first time that GCNs are applied to predict affordances for suction and gripper end effectors in an industrial bin-picking environment. Additionally, we designed a bin-picking oriented data preprocessing pipeline which contributes to ease the learning process and to create a flexible solution for any bin-picking application. To train our models, we created a highly accurate RGB-D/3D dataset which is openly available on demand. Finally, we benchmarked our method against a 2D Fully Convolutional Network based method, improving the top-1 precision score by 1.8% and 1.7% for suction and gripper respectively.

## 1. Introduction

Pick and place are basic operations in most robotic applications, whether in industrial setups (e.g., machine tending, assembling or bin-picking) or in service robotics domains (e.g., agriculture or home). Picking and placing is a mature process in structured scenarios. Nevertheless, it is not the case in less structured industrial environments or when parts with higher degree of variability have to be manipulated. The market demands more flexible systems that will allow for a reduction of costs in the supply chain, increasing the competitiveness for manufacturers and bringing a cost reduction for consumers. The introduction of robotic solutions for picking in unstructured environments requires the development of flexible robotic configurations, robust environment perception, methods for trajectory planning, flexible grasping strategies and human-robot collaboration. Such robotics solutions have not reached the market yet and remain as laboratory prototypes due to the lack of efficiency, robustness and flexibility of currently available manipulation and perception technologies.

The grasping point detection is one of the robotic areas that most attention has attracted since early years of robotic manipulation. The problem of detecting an appropriate grasping point consists of looking for the best object picking location (position and orientation) depending on the type of end effector available. The complexity of the grasping point detection depends on the arrangement of the objects to be handled, and the casuistry varies depending on the structure of the arrangement:Structured: the parts to be picked are organized inside the bin and follow an easily recognizable pattern.Semi-structured: some predictability exists in the way parts are organized and the items are relatively well separated.Random: the parts are randomly organized inside the bin and they do not follow any pattern.

Noticeably, the less the structure of the arrangement, the higher the complexity of the grasping point detection.

The grasping point detection has been traditionally tackled as a computer vision detection problem. In the past, hand-designed features were used to first recognize the object and then to perform model based grasp planning. However, although those methods are robust with small subsets of known objects, they lack of flexibility and are time-consuming. Unlike some other traditional computer vision problems, these detection methods are used on-line and to meet cycle-time restrictions, computational speed is required. The work presented here focuses on flexible, safe and dependable part-handling in industrial environments. In such dynamical areas as warehouses or distribution centres, a huge number of items need to be handled, usually in unstructured scenarios [1].

Traditional grasping point detection methods, even though being efficient in very specific tasks, have proven to be inefficient in tasks with uncertainty and more flexible solutions are demanded. As stated by Kober and Peters in [2], hard coded behaviours are restricted to the situations that the programmer considered and are impractical in dynamic scenarios.

Recently, Deep Learning (DL) based methods have gained popularity due to the affordable price and increasing computation capability of devices such as GPUs and TPUs. DL based methods have shown state-of-the-art performance in a wide range of computer vision, audio and natural language processing problems and it has also been successfully applied to detect grasping points in more complex scenarios [3]. First approaches applied DL to identify randomly placed objects in the scene, to later heuristically obtain grasping points, based on the 3D information of the identified object [4]. However, contemporary approaches do not rely on the identity of the objects and apply DL to predict object affordances. The affordance-based grasping point detection has gained attention due to its capability to learn from the shape, colour and texture of the objects, without relying on part identity. Thus, those algorithms are able to generalize to never seen objects, providing flexible grasping solutions [5].

Yet, most of the DL based grasping point detection algorithms predict grasping points in 2D data, typically images obtained from both RGB or RGB-D cameras, losing the 3D spatial information. Recent approaches use Graph Convolutional Networks (GCNs) to learn geometric features directly in point clouds, and have been successfully applied to predict grasping points [6,7]. Although these methods suffer from computational cost due to the spatial operations applied to the input data, they have shown improved learning capabilities in non-Euclidean data. To alleviate the computational cost, the input clouds are usually sub-sampled both to meet hardware restrictions (e.g., GPU and memory) and to get reasonable computation speeds. In addition to the spatial coordinates of the points in the cloud, these algorithms are able to learn from *n*-dimensional points and, therefore, spatial features such as surface normals can be included in the training data.

The main goal of the work presented here is to analyse whether the usage of *n*-dimensional point clouds and GCNs contributes to better learn object affordances for suction and gripper in a random bin-picking application. The contributions of the paper are as follows:(1)We designed a method based on GCNs to predict object affordances for suction and gripper end effectors in a bin-picking application. This method includes, an adaptation to a point cloud scene segmentation algorithm based on GCNs to predict affordance scores in *n*-dimensional point clouds, and a bin-picking oriented data preprocessing pipeline.(2)We created a highly accurate dataset of random bin-picking scenes using a Photoneo Phoxi M camera, which is openly available on demand.(3)We benchmarked our GCN based approach with the one presented in [5], which uses 2D Fully Convolutional Networks to predict object affordances.

The rest of the paper is structured as follows: Section 2 reviews the literature. Section 3 presents the details of the generated dataset and introduces the 3D affordance grasping approach. Additionally, the evaluation procedure and the used metrics are also detailed. The details of the implementation are explained in Section 4. Finally, the obtained results and the conclusions are presented in Section 5 and Section 6 respectively.

## 2. Literature Review

The problem of finding a suitable grasping point among an infinite set of candidates is a challenging and yet unsolved problem. There are many approaches and a huge variety of methods that try to optimize approximate solutions. According to Sahbani et al. these methods can be categorized as analytic or data-driven [8].

Traditionally, analytical approaches have been used to develop robotic applications to solve specific tasks, mainly implemented with rules based on expert knowledge [9]. Those algorithms are based on kinematic or dynamic models, and the grasping point detection is formulated as a constrained optimization problem [10,11,12]. Even though the analytical approaches are effective, they tend to be very task specific and time-consuming.

Data-driven techniques have proven to overcome the mathematical complexity of the existing analytical methods that solve the grasping point detection problem [13]. These alternative techniques focus more on extracting object representations and are implemented as learning and classification methods. Therefore, grasp representations are learned from experience and used to retrieve correct grasps in new queries. Besides, the parameterization of the grasp is less specific and thus, data-driven algorithms are more robust to uncertainties in perception [14,15,16]. Nevertheless, data-driven methods usually need a big number of annotated data, which usually implies a time-consuming annotation process.

In [13], Bohg et al. split data-driven grasping approaches into three sub-categories, depending on the prior knowledge about the query object:(1)Known objects: The query objects have been previously used to generate grasping experience.(2)Familiar objects: The query objects are similar to the ones used to generate grasping experience. This approaches assume that new objects are grasped similar to old ones.(3)Unknown objects: Those approaches do not assume to have prior grasping experience related to the new object.

When the objects to be manipulated are known and the number of references to be handled is small, a database with 3D objects and predefined grasping candidates is usually used (e.g., created using GraspIt! [17]). First, a pose estimation algorithm is applied to locate the object in the scene, using visual and geometric similarity [18,19,20,21]. Then, grasping candidates are filtered due to collision and reachability issues and the best one is selected, after being ranked by some quality metrics. In spite of being very robust handling one or few object references, such ad-hoc solutions fail when the number of references increases (e.g., due to the unavailability of thousands of 3D models of parts or the computational cost to find such a high number of references in the scene).

However, in industrial setups with highly unstructured environments, changing conditions and highly variable or even unknown multi-reference parts, more flexible methods are needed. To tackle the shortcomings of model based algorithms, most recent approaches use deep neural networks (DNNs) to map robotic sensor readings to labels annotated either by a human or by a robot [22].

In works such as [23,24,25], authors use Convolutional Neural Networks (CNNs) to identify and segment objects in 2D images of the scene. Then, per each identified object, predefined grasping points are used to pick them. Nevertheless, techniques based on the object identification fail to deal with high number of references and novel objects.

Instead of relying on the object identification, other approaches try to predict grasping points focusing on the shape, colour and texture of the parts. In [26], authors created a grasping dataset with both successful and unsuccessful grasps, to later map RGB-D images with graspable regions using CNNs. In [27], the aforementioned dataset was used to create real-time grasps using CNNs. Similar to the previous methods, also in [28,29,30], the grasp was represented as a rectangle over the 2D input images that indicated the point, orientation and the opening of the gripper. In these applications, the grasping pose detector CNN worked in an sliding window manner, performing an inference per detected object in the scene. Thus, the computational complexity depended on the number of objects in the scene. In spite of the fact that the aforementioned methods deal correctly with scattered objects and show good generalization capabilities with never seen objects, they fail to predict grasping points in cluttered scenes [22].

As an alternative to codifying the grasp as a rectangle, other approaches try to predict pixel-wise affordances. For instance, in one of the first attempts, Detry et al. were able to learn grasping affordance models implemented as continuous density functions [31]. In this case, the affordances were learned letting a robot pick and drop objects. However, the learned models were specific to a small set of known objects. Recent methods use Fully Convolutional Networks (FCNs) [32] to learn pixel-wise affordances. Nguyen et al. first applied DL to predict real-time affordances in RGB-D images, to deal with a small set of known objects [33]. In a more recent approach, Zeng et al. used DL to predict pixel-wise affordances for a multifunctional gripper. The predicted grasping scores were used to choose the best action of a predefined set of 4 actions: suction down, suction side, grasp down and flush grasp [5]. Same authors also developed a robotic system that was able to learn to grasp and push objects in a reinforcement learning (RL) setting [34]. In this case, FCNs were used to model affordance based policies. More recently Zeng et al. also made use of FCNs to encode affordance based policies to learn complex behaviours such as picking and throwing objects through the interaction with the environment [35]. In spite of the fact that these FCN based models have shown to be capable to learn object affordances, they can not cope with non-euclidean data such as point clouds. Therefore, the feature extraction is performed uniquely in the RGB/RGB-D images and do not take advantage of the 3D spatial information.

Rather than learning to predict optimal grasping points, in other works authors try to learn visuomotor control policies to directly predict robot actions to pick objects, avoiding the need for an additional robot controller. For instance, Mahler and Goldberg were able to learn a deep policy to pick objects in cluttered scenes in [36]. In [37] authors trained a large CNN to predict the probability of success of task space motions of the gripper, only using RGB images. The learning process was carried out with 14 real robotic manipulators gathering experience in two months of continuous execution. However, the cost of getting experience in reality makes the solution hardly transferable to the industry. Similar to the proceeding proposed by Levine et al., in [38] Kalashnikov et al. used RL to learn high accurate control policies to pick objects in cluttered scenes, also using RGB images. In this approach also the experience of multiple real robots was used to optimize the policy neural network. To avoid the cost of real setups with several robots and to decrease the time to acquire experience, in [39] James et al. proposed to learn the visuomotor control policies in simulation. To reduce the simulation to reality gap, the training process is usually carried out with domain randomization [40].

The aforementioned methods extract grasping features in RGB/RGB-D images, and therefore only a single point of view of the scene is usually used in the learning process. Even if traditional CNNs are able to learn features in euclidean data (e.g., RGB-D images, projections or voxels), they fail to deal with non-euclidean unordered data types such as graphs, were connections between nodes can vary. Recently, Graph Convolutional Networks (GCNs) [41] have gained popularity due to their ability to learn representations over graph-structured data, specially in non-euclidean data such as meshes and point clouds. Similar to 2D CNNs, those models have been used for classification, scene segmentation and instance segmentation [42], particularly in 3D point clouds. As suggested in [43,44], traditional and most used GCNs usually are very shallow models, due to the over-smoothing and over-fitting problem of deep GCNs. Recent works as [45,46,47] try to mitigate these problems introducing several changes in the traditional GCN based model architectures.

GCNs have been successfully applied to solve the grasping point detection problem. Liang et al. used the PointNet [6] network to infer the quality of grasping candidates for a gripper end effector, after applying an antipodal grasp sampling algorithm [48]. Although this technique showed good generalization capability with novel objects, the grasp sampling process was time-consuming due to the need to infer the quality of each proposal. Besides, only local features around the grasp were used to infer the grasp quality, which did not take into account the global object distribution and occlusions that happen in cluttered scenes. Ni et al. proposed a single-shot grasp proposal network for a gripper end effector, based on PointNet++ [7], to avoid the time-consuming grasp candidate sampling [49]. Furthermore, a simulation-based automatic dataset generation proceeding was proposed, using Ferrari and Canny metrics [12]. In spite of the fact that authors were able to predict grasping points also in novel objects, complete 3D models of the objects were used to generate the training dataset in simulation, which are hard to obtain in setups where a big number of references have to be handled. In addition, the grasping dataset was created with single objects, without taking into account global factors in the scene such as occlusions and entanglements. PointNet++ was also used in [50] to implement a single-shot grasp proposal network for a gripper end effector. Although at inference time a single-view point cloud of the scene was used, to train the network, Qin et al. used a simulation based method, that also depends on the availability of 3D models of the parts. In this work also the grasping candidates were generated analytically in single objects and not in cluttered scenes. Although unfeasible grasp proposals are discarded after checking the collisions in cluttered scenes, the global arrangement of the objects is also not taken into account when the grasping dataset is generated.

All the reviewed GCN based methods focused their work only on the gripper end effector and did not consider the grasping point generation for suction. In addition, none of them took into account the global arrangement of the objects when the grasping database was generated. Most of them used simulation to generate the dataset, where it is not straightforward to take into account these global scene factors. Furthermore, none of the reviewed works proposed a bin-picking oriented method.

Generative approaches have also been updated to deal with 3D data, and applied to generate grasping points in point clouds. Mousavian et al. presented a variational auto-encoder which used PointNet++ to implement both the encoder and the decoder [51]. In addition, authors also implemented a PointNet++ based evaluator network to assess the generated grasp candidates. However, this technique dealt with objects with relatively few variability and not in heavy cluttered scenes.

In the presented work, we propose to learn a single-shot affordance-based grasping point detector using GCNs for suction and gripper end effectors, avoiding the need of complex grasp candidate sampling methods. To the best of our knowledge, this is the first time that GCNs are applied to predict affordances both for suction and gripper end effectors in a bin-picking application. Contrary to the general approach in the literature, we propose not to use 3D object models to generate the dataset in simulation, but to use real world scenes instead. Following that idea, we avoid the simulation to reality gap that commonly happens in models trained with synthetically generated data. To annotate the dataset, we take into account global scene factors such as occlusions (e.g., partially occluded objects are graspable or not depending on the weight of the objects that covers them), object entanglements, etc. Those are difficult to take into account in simulation but crucial in real applications. With that purpose, we have created a highly accurate dataset in real random bin-picking scenes using the industrial 3D Photoneo Phoxi M camera [52]. Although such expensive cameras are not widespread for laboratory level applications, the high accuracy they offer is vital in industrial bin-picking setups. Furthermore, we show that GCNs are able to converge with a relatively small training dataset of *n*-dimensional point clouds using data augmentation.

## 3. Problem Specification and Setup

The random bin-picking problem can be divided into the following categories:(1)Mono-reference: the objects are randomly placed but belong to the same reference, which is usually known (e.g., Figure 1a).(2)Multi-reference: the objects are randomly placed and belong to multiple references. The number of references can be high and novel objects may appear in the bin (e.g., Figure 1b).

In mono-reference applications, where the parts to be handled are known and their models can be easily obtained, 3D model matching algorithms are typically used to estimate the 6-DoF pose of the parts in the scene [53]. However, the multi-reference random bin-picking is an unsolved problem yet essential for flexible/efficient solutions.

Our work is focused on the context of multi-reference bin-picking. DL based algorithms have been widely used in the literature to handle uncertainties in the scene and novel objects. Particularly, methods that predict object affordances have shown to be robust to uncertainties and have a good generalization capability. However, most of them are only able to handle 2D data, and do not take into account 3D spatial features of the parts/scene. Recently, Zeng et al. won the Amazon Robotics Challenge (ARC) with a bin-picking method that was based on affordances [5]. Authors faced the multi-reference random bin-picking as a 2D scene segmentation problem and were able to get pixel-wise affordance scores. Based on that work, our intuition was that learning directly in *n*-dimensional point clouds would lead to a performance boost in the grasping point detection.

With that aim, we selected the Deep GCN model proposed in [46] where some novelties were introduced that led to a very deep GCN based model. Although originally it was implemented for segmentation purposes, the adaptation we introduced allows us to obtain affordance scores in bin-picking scenes, following the idea of Zeng et al. In addition, we followed a bin-picking oriented pipeline, which made the learned GCNs applicable to other picking scenarios, as it is later on explained in Section 3.2.2.

For evaluation purposes, we have created a dataset with multiple annotated bin-picking scenes. The dataset was annotated once and we used it to train and test both methods (FCNs and GCNs). The details of the dataset and the annotations are explained in Section 3.2. Finally, in order to make a fair comparison, the metrics defined in [5] have been used to measure the deep models. These are further explained in detail in Section 3.4.

### 3.1. Multi-Functional Gripper

In the context of the Pick-Place European project [54], a multi-functional gripper has been developed by Mondragon Assembly (Figure 2). This multi-functional gripper is composed of the following retractable end effectors: A magnet, a suction cup and a two finger gripper. All of them are equipped with tactile sensors developed by Fraunhofer—IFF that are designed to softly handle any kind of product.

Although in this work the gripper was not directly used to annotate the dataset, it was important to define its particularities since the annotation of the dataset highly depends on these specifications. In this way, the graspability of the objects was tested individually with each end effector. For instance, the quality of a grasping point can drastically change depending on the maximum width between the fingers of the gripper. In this work, both, suction and gripper end effectors were considered.

### 3.2. Dataset

In industrial bin-picking applications, the perception of the scene must be as accurate as possible, due to the precision needed to handle objects with variable shape, colour and texture. At the same time, industrial 3D cameras are becoming more and more robust against changing environmental conditions. In a first attempt, we chose to profit from the dataset provided by [5]. However, the bin localization was not available for the suction and, preliminary results showed that the depth data was not accurate enough due to the location of the camera. Thus, this option was discarded. Instead, we opted to use a Photoneo Phoxi M camera to generate our dataset, which has been widely used in many bin-picking applications and has demonstrated a great performance [55,56]. We positioned the camera overhead the bin to reduce occlusions. An example of a captured scene and its corresponding point cloud are depicted in Figure 3. To generate the dataset, we used a set of 37 rigid and semi-rigid opaque parts that were selected in the context of the Pick-Place EU project. In spite of the fact that we included some transparent and shiny objects, we opted to focus our analysis on opaque parts.

As a result, the dataset was composed of the following elements per each scene:Intrinsic parameters of the camera.RGB-D images of the scene. Using the intrinsic parameters, images can be easily transformed to a single-view point cloud.Bin localization with respect to the camera.Point cloud of the scene.

As RGB-D images can be easily converted to single-view point clouds, the annotation process was carried out only in RGB-D data.

#### 3.2.1. Annotation

The performance of the learned models highly depends on the quality of the annotated data. Although simulation based dataset annotation methods are widely used, we believe that the experience and the criteria of the human is vital to analyze the particular casuistry of each scene. To overcome this burden, we created a small dataset composed of 540 scenes with randomly placed multi-reference parts. We followed the annotation style proposed in [5] and we labeled the acquired scenes twice, for suction and gripper end effectors respectively. The labeling process was made taking into account the restrictions of the multi-functional gripper presented in Section 3.1. We physically used the multi-functional gripper to extract the good and bad grasping areas in individual parts and applied this knowledge to manually annotate complex scenarios. The manual annotation lets us take into account weight distributions, object entanglements and complex situations in general that are difficult to consider in simulation. In the application presented by [5], authors tackled the grasping point detection as a scene segmentation problem, being able to predict pixel-wise affordances. For that purpose, the RGB-D dataset was annotated pixel-wise. The main particularities of each type of annotation are described below:

##### Suction Annotations

For the suction end effector, a grasping point is defined by a 3D contact point in the scene along with its normal vector. Here, the normal vector determines the orientation of the suction end effector. As depicted in Figure 4a, the pixels that belong to objects and are affordable for the suction tool, were annotated with green colour. The pixels that belong to objects but are not good grasping points, were annotated with red colour. Finally, the rest of the points that belonged to the scene were annotated as neutral. Suction annotations were stored as masks, and the value of each pixel of the mask indicated the class of the corresponding pixel in the original image. The annotation process was carried out using the Pixel Annotation Tool [57].

##### Gripper Annotations

A gripper’s grasping point is defined by:(1)A 3D point in the space that indicates the position where the center between the fingers of the gripper has to be placed.(2)The orientation of the gripper in the vertical axis.(3)The opening of the fingers.

The RGB-D scenes were transformed into orthographic heightmaps, and top views of the scenes were obtained, due to the fact that only vertical gripper actions were taken into account. For that purpose, first we located the bin with a 3D model matching algorithm, and we transformed the clouds into the bin coordinate frame. Then, the points that laid outside the bin were discarded and the resulting cloud was projected orthographically and stored again as RGB-D image.

As it can be seen in Figure 4b, grasps were annotated with straight lines. Good and bad grasps are represented with orange and blue colours respectively. Following that approach, only hard negative grasping points were annotated as bad. The center of the line indicates the 3D point in the space where the gripper should move to, and the orientation of the line with respect to the horizontal axis of the image indicates the grasping angle. The opening of the gripper fingers is computed on-line during execution, taking into account the local geometry of the grasping area.

Similar to the annotation proceeding proposed in [5], the annotations made with lines were converted to pixel-wise labels. We only took into account rotations in the *z* axis to only perform vertical grasps. For the sake of simplifying the problem, the *z* axis was divided into *n* fixed angles, in our case n=16. To decide to which discrete angle each annotated line belonged to, the angle θ between each line and the horizontal axis of the heightmap was computed. Thus, for each annotated RGB-D scene for the gripper, 16 pixel-wise annotations were obtained (one mask per angle). The transformation from the line annotations into masks was done in the following way:(1)The θ angle between each line and the horizontal (*y*) axis indicates the discrete *n* orientation in *z* axis that the annotation belongs to.(2)The scene RGB-D image is rotated *n* times with an increment of 360/n with respect to the *z* axis of the bin.(3)The center pixel of each annotated line is computed and included in the corresponding annotation mask among the *n* possible orientations.

To annotate the dataset for the gripper, we modified the VGG Image Annotator to take into account the angle of each annotation with respect to the horizontal axis of the heightmap (*y*) [58].

#### 3.2.2. RGB-D Annotations to 3D Point Clouds

In this section are detailed the steps carried out to transform RGB-D annotated data into 3D annotated point clouds for both, suction and gripper effectors.

##### Suction

Our goal was to follow traditional preprocessing pipelines proposed in bin-picking applications, with the next requirements: To ease the learning process of grasping points in 3D point clouds and to create a generic solution for any bin-picking application. For that purposes, we designed the pipeline showed in Figure 5 and afterwards we used it to transform the RGB-D dataset into a single-view point cloud dataset. Basically, we aimed to abstract from the camera pose in the scene by always working in the bin coordinate frame. This also gives the possibility to work with multiple extrinsically calibrated cameras.

2 Conforming to the pipeline, first, the RGB-D images were transformed into point clouds using the camera intrinsics matrix. Then, the pose of the bin was obtained, using a 3D model matching algorithm previously developed at Tekniker [1]. As the dimensions of the bin were known, the bin localization algorithm created the 3D model of the top edges of the bin and matched it to the obtained point cloud of the scene, to finally obtain its pose. For that purpose the Mvtec Halcon [59] library was used. Finally, the clouds were transformed into the new coordinate frame and the points that laid outside the bin were filtered using the bin size. In this case, all the tests were made with a bin size of length=600 mm, width=400 mm and height=230 mm. Additionally, the standard voxel downsampling algorithm offered by the Open3D [60] library was used, with voxel size 0.002 m, followed by a statistical outlier removal from the same library.

Alongside the training data, also the annotations needed to be transformed into 3D. Figure 6 shows the original (Figure 6a) and the transformed (Figure 6b) annotations for the suction end effector. The output of the transformation was a semantically annotated 3D point cloud where: green, blue and pink pixels belong to good, bad and neutral grasping point classes, respectively. In this case, the annotation transformation into 3D was straightforward, and point classes were assigned depending on which class the original RGB-D pixel belonged to.

##### Gripper

Contrary to the followed procedure for the suction, in this case, the annotated RGB-D orthographic heightmap was already transformed into the bin coordinate frame and, thus, the conversion of RGB-D images to 3D point clouds in the bin coordinate frame was straightforward. As well as from each annotated RGB-D image a labeled point cloud was obtained for suction, in the case of the gripper, instead, *n* point clouds were obtained. As explained before, we only took into account rotations in the *z* axis to only perform vertical grasps. The transformation from 2D annotations into 3D was made in the following way:(1)The θ angle between each line and the *y* axis (horizontal axis of the orthographic heightmap) indicates to which discrete *n* orientation in the *z* axis the annotation belongs to.(2)The scene point cloud was rotated *n* times with an increment of 360/n with respect to the *z* axis of the bin.(3)The 3D coordinate of the centre pixel of each annotated line was computed in the corresponding point cloud among the *n* rotated clouds.(4)The length of each line indicates the radius of the circumference around the centre pixel that was annotated.

An example of the gripper annotation conversion to 3D can be seen in Figure 7 and Figure 8. On the one hand, the annotated RGB-D orthographic heightmap can be seen in Figure 7. On the other hand, two of the 16 generated annotated clouds are depicted in Figure 8. Specifically, n=6 and n=12 discrete orientations in the *z* axis have been selected for illustration purposes. In these examples the *y* axis (green) indicates the orientation of the gripper.

### 3.3. 3D Affordance Grasping with Deep GCNs

In spite of the fact that CNNs have shown strong performance with Euclidean data, this is not the case for many applications that deal with non-euclidean data. Graphs are popular data structures that are used in many applications such as social networks, natural language processing, biology or computer vision [61]. Although traditional CNNs are not able to systematically handle this kind of data, GCNs have shown to be able to overcome the shortcomings of CNNs.

A graph G=(V,E) is defined by an unordered set of vertices V and a set of edges E indicating the connections between vertices. Vertices are represented associating each vertex v∈V with a feature vector $h_{v}\in R^{D}$, where D indicates the dimension of the feature vectors. So, the graph is represented concatenating the feature vectors of the set of unordered vertices $h_{G}=\left[h_{v_{1}},h_{v_{2}},h_{v_{3}},...,h_{v_{N}}\right]\in R^{DxN}$, where *N* indicates the number of vertices in the graph.

In GCNs, the most used operations at each layer are aggregation and update. These operations receive the input graph $G_{l}=\left(V_{l},E_{l}\right)$ and they output the graph $G_{l+1}=\left(V_{l+1},E_{l+1}\right)$ at the *l*-th layer. On the one hand, the aggregation function is used to collect the information of the neighbour vertices (e.g., max-pooling aggregator). On the other hand, the update function applies a transformation to the collected data by the aggregate operation, in order to generate new vertex representations (e.g., using MLPs). At each layer, the aforementioned operations are applied to each vertex $v\in V_{l}$ and new vertex representations are generated [62].

Most GCNs have fixed graph structures and only per-vertex features are updated at each iteration. However, more recent works demonstrate that changing graph structures contribute to better learn graph representations. For instance, Wang et al. proposed a neural network module dubbed EdgeConv [63] that finds *k* nearest neighbours in the feature space to reconstruct the graph at each EdgeConv layer. The authors claim that this architecture is suitable for classification and segmentation tasks in point clouds.

Furthermore, as reported in recent work, traditional GCNs cannot go as deep as CNNs due to the high complexity in back-propagation and they are no more than three layers deep [43,44]. State-of-the-art works suggest multiple changes in GCN architectures to overcome the aforementioned shortcomings [45,46,47]. The Deep GCNs model architecture developed by Li et al. in [46] proposes multiple changes that allow to effectively learn deep GCNs that are up to 56 layers deep. Moreover, this architecture has shown great performance in semantic segmentation tasks in the S3DIS indoor 3D point cloud dataset [64].

The adaptations that allow deep GCNs are twofold:Dilated aggregation: Dilated *k*-NN is used to find dilated neighbours after every GCN layer, getting as result a Dilated Graph. Having a graph G=(V,E) with a Dilated *k*-NN and *d* dilation rate, the Dilated *k*-NN returns the k×d nearest neighbours in the feature space, ignoring every *d* neighbours. The $l_{2}$ distance is used in the feature space.Residual and Dense connections: Based on the success of models with residual connections between layers such as ResNet [65], or dense connections such as DenseNet [66], these concepts have been translated to GCNs, allowing much deeper models.

The model chosen for our affordance-based grasping algorithm includes the dilated aggregation so to increase the receptive field of the GCN. Additionally, we chose the residual connections between the layers to increase the depth of the network. The selection of this network configuration was motivated by the improved performance achieved against other architectures in [46]. The used architecture is illustrated in Figure 9.

When the model is fed with the *n*-dimensional point cloud data, first the GCN backbone is in charge of extracting features from the input data. Then, the fusion block fuses the extracted features by the backbone and global features are extracted. Finally, the prediction block predicts point-wise labels. As it can be seen in Figure 9, we modified this latter block to obtain point-wise affordance scores. The output tensor of the network has the following shape: $\left(N_{batch},N_{pts},N_{classes}\right)$, where $N_{batch}$ indicates the number of batches fed to the network, $N_{pts}$ is the number of points per batch, and $N_{classes}$ the number of classes. In our case $N_{classes}=3$ (good, bad and neutral grasping points). Thus, to obtain the point-wise labels, the *argmax* operation is performed over the last axis of the output tensor. To obtain good grasping affordances, however, we only get the channel corresponding to good grasping points, as the *softmax* operation outputs the probability distribution over the three classes. In our case the affordances are predicted in the first channel of the last axis, as shown in Equation (Equation 1).
(1)$$affordances=[N_{batch},N_{pts},0]$$

Doing so, the points can be ordered depending on their affordance score. The higher the affordance score, the higher is the likelihood of this grasp being a success. We used the same general model architecture both for suction and gripper end effectors. Nevertheless, we selected specific hyper-parameters for each case, as it is later explained in Section 4.

### 3.4. Benchmark Definition and Metrics

Here we describe the tests defined to compare both FCN and GCN based methods and the metrics defined to measure the performance of each of them.

#### 3.4.1. Benchmark Definition

To assess each model and to make a comparison between them, two test were carried out:(1)Test 1: The methods were trained with the 80% and assessed against the remaining 20% of the dataset, that was composed of scenes with known objects, but completely new object arrangements that were not used to train the model.(2)Test 2: We created a set of 100 new scenes containing randomly arranged similar but never seen objects, in order to assess the generalization capability of each model. To that end, more than 15 new parts were selected.

#### 3.4.2. Metrics

As authors claim in [5], a method is robust if it is able to consistently find at least one suction or grasp proposal that works. Thus, the metric used to measure the robustness of the methods is the *precision* of the predictions against the manual annotations. The precision was computed with respect to multiple confidence levels:Top−1 precision: For each scene, the pixel (for the FCN) and the point (for the GCN) with highest affordance score was taken into account to measure the precision.Top−1% precision: In this case the pixels/points were sorted according to their affordance scores and those within the 99th percentile were selected to measure the precision.

In both cases, a grasping proposal is considered as a true positive if it has been manually annotated as good grasping area, and as a false positive if has been manually annotated as bad grasping area.

## 4. Implementation

In this section all the details related to the data preprocessing, network configurations and used training hyper-parameters are explained.

### 4.1. Data Preprocessing for the Suction

In order to fit the generated 3D dataset with the GCN model for the suction, some preprocessing steps have been performed. First, the number of points that GCNs could analyze was limited due to computational complexity issues and hardware restrictions. In spite of the fact that the general approach in the literature is to split the input point cloud in blocks later to sequentially feed the model (using a reduced number of points per block), we decided our model to be a single-shot detector. Therefore we increased the number of points to be processed in each batch and set it to 8192. As proposed in [46], we used a random sampling method to reduce the sampling time compared to other complex sampling methods. For suction, each input point was defined by a 9-dimensional feature vector containing, (see Equation (Equation 2))

(1)The point coordinates with respect to the bin coordinate system.(2)RGB values normalized to [0–1].(3)Normal vector computed using the points within a radius of 0.05m before the random sampling.

(2)$$\left[x_{bin},y_{bin},z_{bin},r,g,b,n_{x},n_{y},n_{z}\right]$$

Here our intuition was that the usage of normal vector information (not available in 2D) would lead to a performance boost, as most suction grasping points are located in flat and regular surfaces. Furthermore, as the annotated data was limited, we applied some data augmentation to generate more training data. For suction, after randomly sampling 8192 points from the scene, the sampled point cloud was randomly rotated in the *z* axis. As good grasping points do not change when the input cloud is rotated with respect to the vertical axis, the augmentation was straightforward.

### 4.2. Data Preprocessing for the Gripper

Similar to the followed preprocessing steps for the suction, again 8192 points were randomly sampled for each scene. As explained in Section 3.2, for each annotated heightmap, as many rotated point clouds as discrete angles *n* were obtained. Thus, *n* inferences were performed to detect grasping points in a scene for the gripper. Each input point was defined by a 6-dimensional feature vector (Equation (Equation 3)):(1)The point coordinates with respect to the bin coordinate system.(2)RGB values normalized to [0–1].
(3)$$\left[x_{bin},y_{bin},z_{bin},r,g,b\right]$$

As gripper grasping points usually are in irregular surfaces, our intuition was that normal vector information would not contribute to better learn grasping point representations. Thus, we opted to exclude it. Aside from that, in this case it was not possible to augment the data rotating it vertically, as the rotation of the data had implicit information of the grasping orientation. Therefore, we considered the random sampling as data augmentation, as each sampled point set was totally different to the previously generated ones.

### 4.3. Training

The details of the tuning hyper-parameters used in our tests are shown in Figure 10. The parameters that are not mentioned, were left as default. As aforementioned in Section 3.2, both for suction and gripper, the dataset was annotated using three classes: Good, bad and neutral. However, as the main goal was to distinguish between good and bad points, and to equalize the instances belonging to each class, the background class was trained with zero loss. Therefore, the elements belonging to that class were not taken into account when the loss was computed. The random sampling and the data augmentation were applied on-line at each training step. We trained our models in the cloud using an AWS EC2 instance with a *16 GB Nvidia V100* GPU.

## 5. Results

In this section are detailed the results obtained in the tests defined in Section 3.4. In each and every performed test, in addition to the obtained precision scores, a graphical example was given with the following colour/class correspondence: green for good grasping points, blue for bad grasping points and pink for neutral points.

### 5.1. Test 1

In this test the algorithms were trained with the 80% of the dataset and assessed against the remaining 20%. The metrics defined in Section 3.4 were used to compare both methods.

#### 5.1.1. Suction

The results obtained with both suction FCN and GCN models are shown in Figure 11. In the case of the FCN, as the 2D input data were constant for testing, it was executed once. Nevertheless, as the *n*-dimensional input points were randomly sampled and with the aim of measuring the average performance, the GCN was executed five times. Therefore, the precision scores obtained with the GCN slightly varied.

The results showed that the deep GCN was able to outperform the precision scores obtained by the FCN. On the one hand, the usage of *n*-dimensional spatial data helped to better learn grasping point representations for the suction. As most of the suctionable areas were located in flat surfaces, the introduction of the normal vectors in the learning process led to a performance boost. On the other hand, the data augmentation played a key role, as the annotated data was somehow limited. First, the random sampling avoided over-fitting the input data, since the obtained point set after each sampling iteration was totally different to the previously used ones to train the model. In addition, the random rotation in *z* axis of the sampled point set, contributed even more to generate new grasping points that helped to learn more generic grasp representations.

Nevertheless, the data augmentation was not trivial for the 2D FCN. Although visual transformations could be applied to the RGB image (light, texture, colour, ⋯), it was not trivial to augment the depth data, as it was stored as a 2D image and had pixel-wise correspondence with the RGB image. In addition, it was not clear up to what extent 2D models as FCNs take advantage of the 3D spatial information, since traditional 2D convolution operations were designed to only extract 2D features. As our training dataset was not so big, the FCN over-fitted it in few training iterations and, thus, the model was not able to learn such good grasp representations as GCNs.

The designed bin-picking oriented pipeline also contributed to the improvement of the results. As all scenes were transformed into the bin coordinate frame and points outside the bin were discarded, the learning problem was simplified and the GCN showed an improved performance, independently of the location of the bin.

An example of an inference of the GCN with a scene obtained from the test split of the dataset is depicted in Figure 12. For illustration purposes, top-1% and top-5% precisions have been selected.

#### 5.1.2. Gripper

The results obtained for the gripper with both, FCN and GCN models, are depicted in Figure 13. We followed the same methodology as in the case of the suction, with one and five executions for the FCN and GCN respectively. Since in the GCN based approach points were sampled from the original scene, contrary to the FCN based method, the obtained precisions vary. As it can be seen in Figure 13a, the GCN based method outperformed the results obtained with the FCN, giving at least a valid grasping point per each scene with high precision. Nevertheless, Figure 13b shows that the FCN performed more precisely taking into account the highest 1% of the predicted affordances. This means that the GCN showed a more overconfident performance than the FCN, sometimes assigning high affordance scores to points that did not deserve it.

On the one hand, the random sampling of the input data led the GCN model to learn generic features and prevented it from over-fitting the input data. On the other hand, the fact that the GCN learned from *n*-dimensional features allowed the model to be more confident predicting top-1 affordances. In spite of the fact that the GCN was more confident taking into account the maximum affordance value per each scene, this confidence dropped when the affordances in the 99th percentile were taken into account and the FCN showed a stronger performance.

The results obtained with two different vertical angles for a single scene are shown in Figure 14 and Figure 15. For illustration purposes, the n=4 and n=5 discrete vertical angles have been selected, among the n=16 possible rotations.

In these examples it can be appreciated that, taking into account the top-1% affordance scores, the GCN model was able to also correctly predict grasping points also in objects that were not annotated as good or bad. In addition, sometimes the model was overconfident and predicted relatively high affordance scores to points that were not good grasping points. This could happen due to the fact that only hard negative samples are annotated as bad. Nonetheless, in both examples the predicted top-1 grasping points are correct and the top-1 precision of the GCN supports its capability to learn grasping points for the gripper.

### 5.2. Test 2

The main goal of this test was to assess the generalization capability of the trained models for suction and gripper. For that purpose, the previously trained models were assessed against a set of 100 new scenes, composed of randomly arranged completely new parts.

#### 5.2.1. Suction

The achieved top-1 and top-1% precision scores for the FCN and the GCN are depicted in Figure 16. Similar to the previous tests, also in this test the same number of executions were performed per each model. In spite of the fact that both models behaved precisely when they were assessed with scenes full of never seen objects, the GCN showed a stronger generalization capability, obtaining higher top-1 and top-1% precision scores.

Regarding the generalization capability of the FCN, although no data augmentation methods were applied, in few iterations it learned generic enough features to correctly predict affordances in new objects with random arrangements. Nevertheless, the data augmentation and the random input data sampling had a lot to do with the results obtained with the GCN. Due to the fact that all point clouds were transformed into the bin coordinate frame, the application of these preprocessing steps was straightforward. Consequently, this preprocessing prevented the model from over-fitting the input data, being able to learn more meaningful spatial features than with the 2D FCN.

A graphical example of the predicted affordances with top-1% and top-5% confidences are shown in Figure 17. Watching this graphical results, it can be said that the learned GCN model shows a strong capability to predict affordances for randomly placed completely new parts.

#### 5.2.2. Gripper

As it can be seen in Figure 18, the precision of both, FCN and GCN decreased considerably when the models were assessed in scenes with totally new objects. Particularly in the case of the GCN, the performance worsened even more than in the case of the FCN, suggesting that the learned model focused more on learning specific features of the training data.

For the gripper, the random sampling was considered as a data augmentation method, due to the fact that the input data varied at each training step. Nonetheless, it was not enough to learn generic grasp representations for the gripper. Contrary to the case of the suction where the GCN model was able to correctly extrapolate how to predict affordances in new parts, in the case of the gripper it seemed that the model focused too much in the particularities of the training objects. In Figure 19 and Figure 20, the affordances obtained for two different vertical angles for the gripper are shown, for n=2 and n=3 respectively.

In these graphical results we can see that, in some cases the model assigned high confidence values to points that were not annotated as good and, indeed, were not good grasping points. As an example, in spite of the fact that the top-1 grasping point predicted in Figure 19 had the correct orientation, it was not an adequate grasping point for the gripper due to the width of the part. In some other cases as in Figure 20, however, the model found a correct top-1 grasping point although it assigned high affordances also to points that did not deserve it.

## 6. Discussion

In the work presented here we used a GCN to test if the assumption that *n*-dimensional information is crucial for predicting object affordances fulfills. We successfully adapted the *Deep GCN* model that was designed for scene segmentation problems to predict object affordance scores for suction and gripper end effectors. To train the GCNs, we created a dataset composed of industrial bin-picking scenarios with randomly arranged multi-reference parts. To that end, we used a Photoneo Phoxi M camera and obtained highly accurate 2D/3D acquisitions of multiple scenes. In our application we selected a varied set of rigid and semi-rigid objects with different material, shape, colour and texture that were used in the context of the Pick-Place EU project.

Rather than 2D images containing single views of the scene, we used *n*-dimensional point clouds offering a richer information of the scene (e.g., multiple viewpoints and multiple features per point). Although the usage of unordered *n*-dimensional point clouds introduced complexity to the learning process, we showed that it is possible to learn to predict object affordances with GCNs. The innovations introduced in [46] let us create deep GCN models for suction and gripper end effectors. Besides, the designed data processing pipeline contributed to create a system which was agnostic to the bin localization in the scene and to the number of cameras being used. Thus, the designed methodology was easily transferable to new scenarios and setups.

Traditionally, DL based methods need arduous manual annotation processes which sometimes make those applications intractable. This drawback of DL applications highlights the importance of automatic data augmentation methods, to automatically increase the training samples in the dataset. Synthetically augmented datasets are widely used and help to generate much data with little effort. However, simulation based methods increase the simulation to reality gap, due to the difficulty to replicate real world conditions in simulators. We directly augmented real *n*-dimensional spatial data. As we are using 3D data the process was rather trivial contrary to augmenting RGB images where data is arranged in a grid.

The quantitative performance measures obtained confirm our assumption and we can say that indeed 3D spatial information contributes to predict object affordances more precisely. Test 1 showed that the GCNs correctly predict object affordances with known objects but in completely new scenarios and arrangements, obtaining better precisions than the 2D FCN. On the other hand, Test 2 allowed us to check the generalization capability of the trained models. These models had to predict affordances in new objects with random arrangements. The obtained results demonstrated that the GCN based suction model had strong generalization capabilities to correctly predict affordances in similar but completely new parts. However, the precision scores obtained with the gripper indicate that the FCN generalized better to new parts than the GCN, suggesting that the input data was not significant enough for the GCN in the more complex gripper scenario.

Despite the promising results obtained predicting object affordances by training GCNs with *n*-dimensional spatial features, the system suffered from various limitations. The former was directly related to the resolution of the data. As each point in the scene is represented as a *n*-dimensional vector, the computational cost increases proportionally when more features are included. Due to hardware and cycle time restrictions, the amount of data to be processed at each step is limited. In this work, each scene was represented by 8192 points, which seemed to be enough in our case with relatively big objects, but not with very small parts. Even though the general approach with GCNs is to split the point cloud in smaller cubes to gain resolution, the general perspective of the scene is lost, which is crucial in the grasping point detection problem.

The second limitation came when the system had to deal with transparent or shiny parts. As most of the depth sensors fail to reconstruct the 3D information of these parts, typically only visual information can be used to infer their affordances.

As far as the grasping strategies are concerned, the developed method for suction takes full advantage of the 3D space and grasps are predicted and executed with 6-DoF. However, that is not the case for the gripper, where the affordances are only predicted taking into account vertical grasps and discrete angles. Consequently, the computational cost is proportional to the selected number of discrete angles, and that makes the solution hardly scalable.

The developed work allowed us to gain experience and knowledge that should be enriched by testing the models with the real robotic system in a real application to see whether the learned grasping representations are valid to pick real objects. Moreover, the manual annotation is a very time-consuming process. Although data augmentation techniques somehow alleviate it, the DL models still need too much annotated data to converge. Thus, we must look for ways to learn to predict object affordances, with less manually annotated samples. In spite of the fact that current state-of-the-art GCN based approaches mostly focus only on the gripper end effector, it would be enriching to benchmark our work with other methods in a bin-picking scenario. Lastly, the grasping strategy for the gripper must be extended to 6-DoF in a more flexible way to overcome the current generalization and scalability limitations.

## Figures and Tables

**Figure 1 sensors-21-00816-f001:**
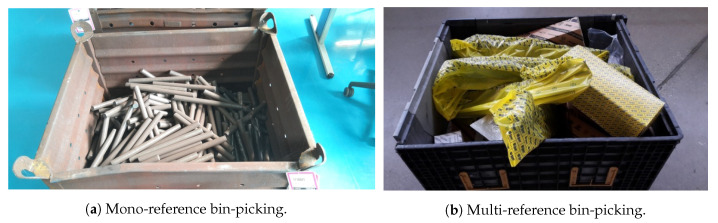
Random bin-picking scenarios.

**Figure 2 sensors-21-00816-f002:**
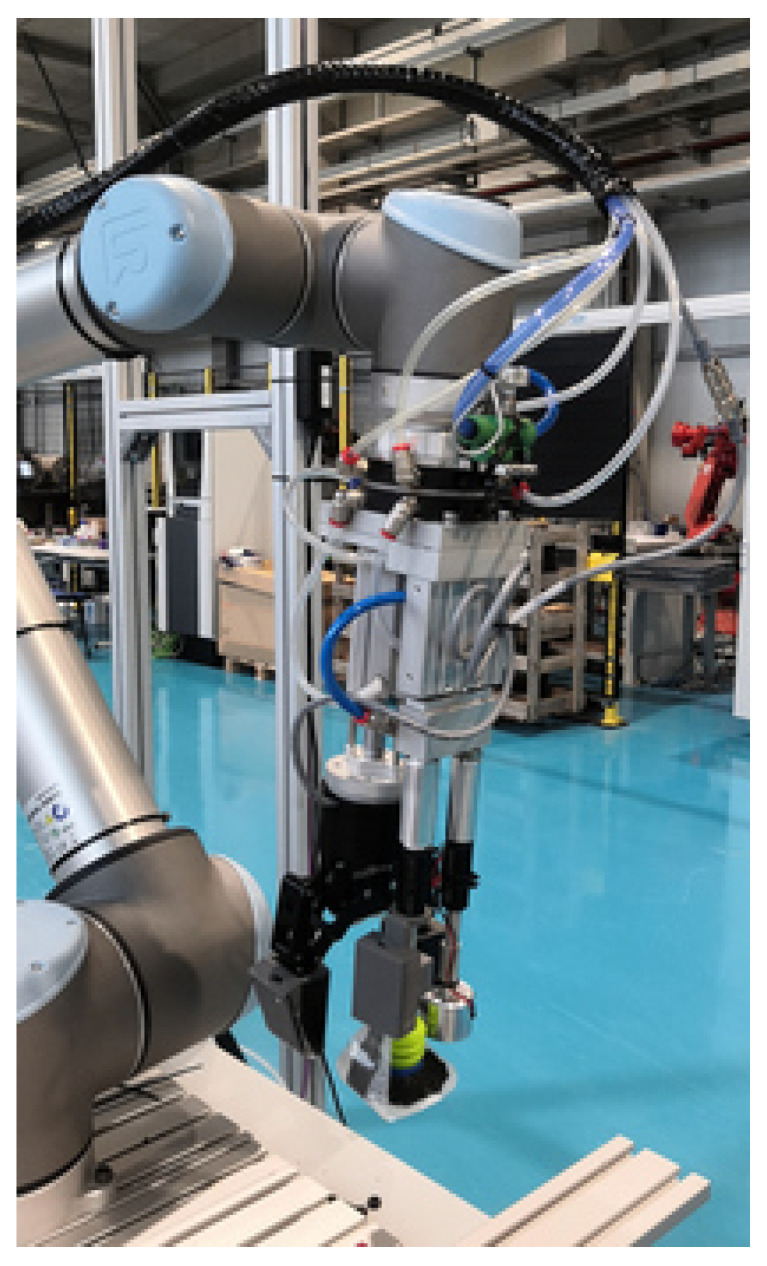
Multi-functional gripper attached to the UR robot.

**Figure 3 sensors-21-00816-f003:**
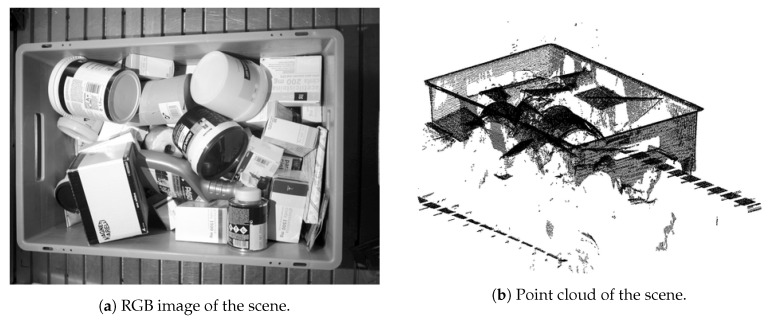
Dataset sample scene.

**Figure 4 sensors-21-00816-f004:**
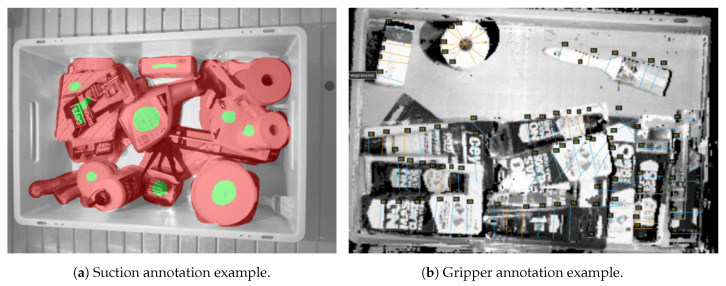
Annotation examples in RGB-D data.

**Figure 5 sensors-21-00816-f005:**
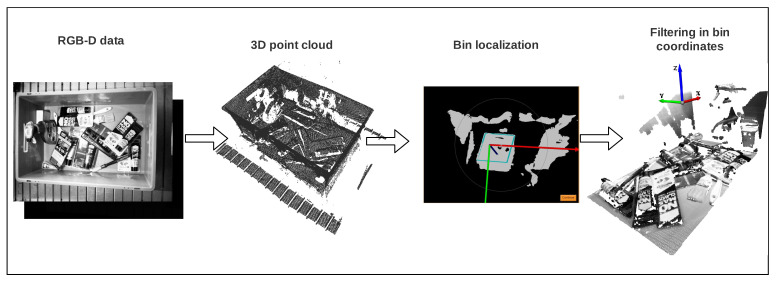
Data preprocessing pipeline.

**Figure 6 sensors-21-00816-f006:**
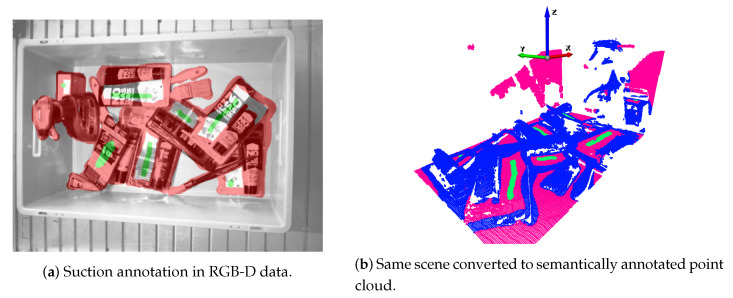
Annotation transformation for the suction.

**Figure 7 sensors-21-00816-f007:**
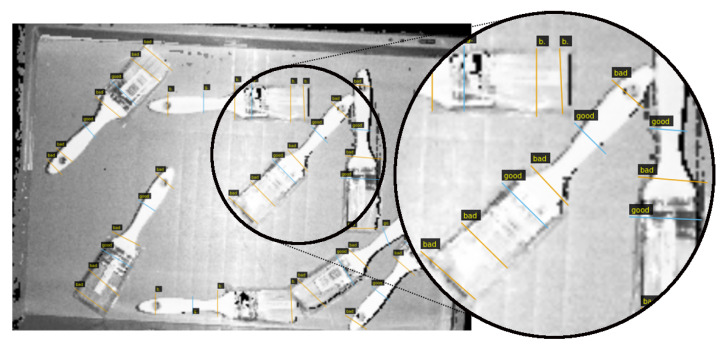
Gripper annotations in a RGB-D orthographic heightmap.

**Figure 8 sensors-21-00816-f008:**
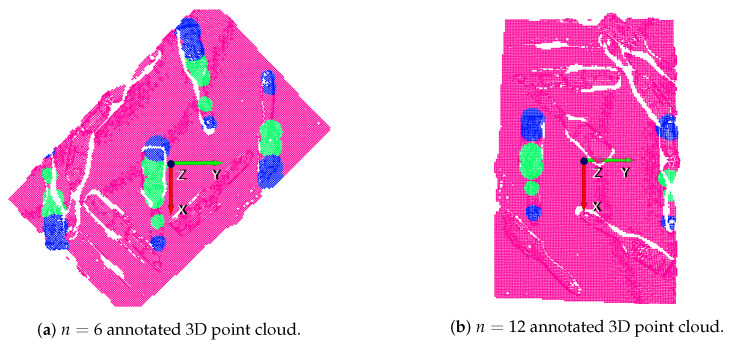
Annotation transformation for gripper.

**Figure 9 sensors-21-00816-f009:**
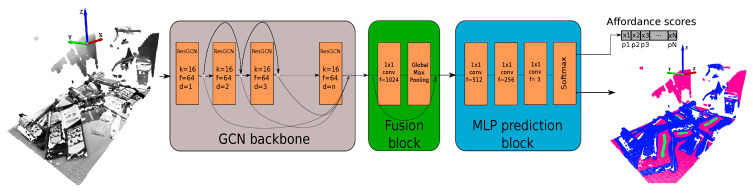
Used model architecture.

**Figure 10 sensors-21-00816-f010:**
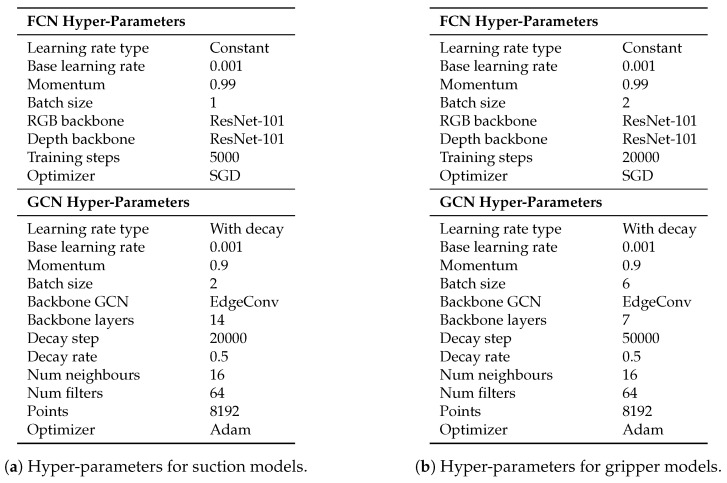
Hyper-parameters for suction and gripper Fully Convolutional Network (FCN) and Graph Convolutional Network (GCN) models.

**Figure 11 sensors-21-00816-f011:**
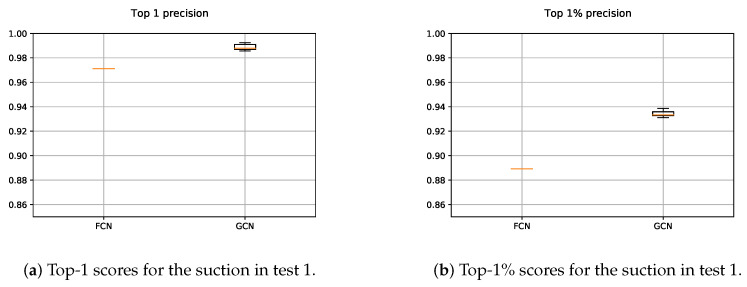
Obtained precision scores per confidence percentiles for the suction models in Test 1.

**Figure 12 sensors-21-00816-f012:**
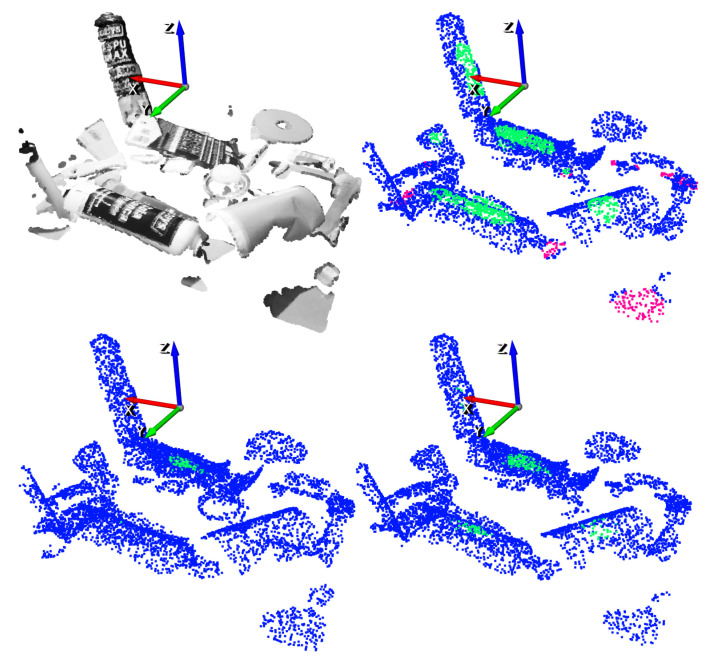
Suction result example. Top-left: Point cloud of the scene. Top-right: Ground-truth annotation. Bottom-left: top-1% predictions. Bottom-right: top-5% predictions.

**Figure 13 sensors-21-00816-f013:**
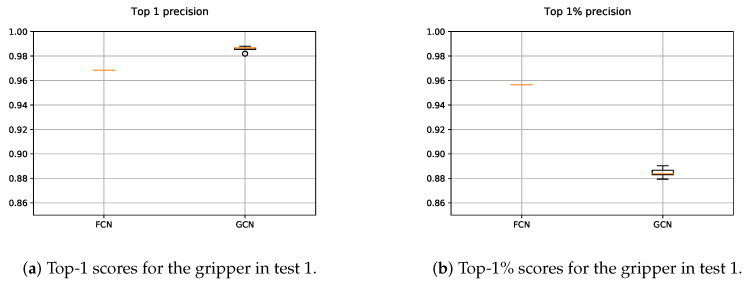
Obtained precision scores per confidence percentiles for the gripper models in Test 1.

**Figure 14 sensors-21-00816-f014:**
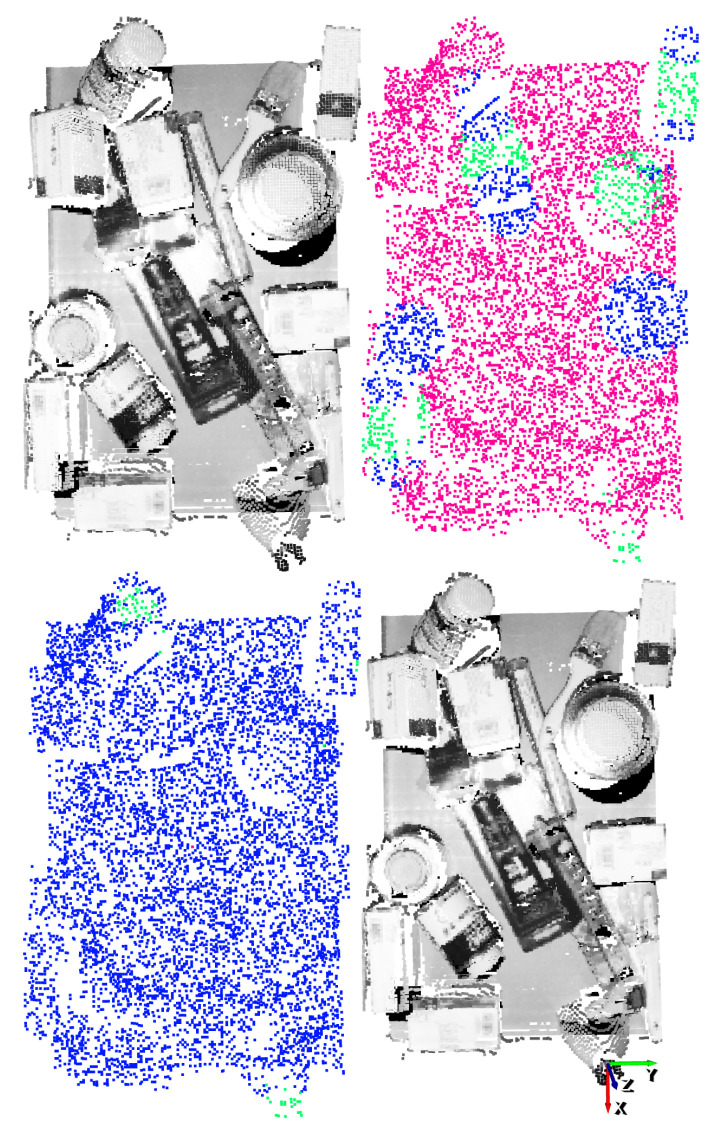
Gripper result example for n=4. Top-left: point cloud of the scene. Top-right: ground-truth annotation. Bottom-left: Top-1% predictions. Bottom-right: Top-1 grasping point. The vertical orientation for the gripper is determined by the *y* (green) axis.

**Figure 15 sensors-21-00816-f015:**
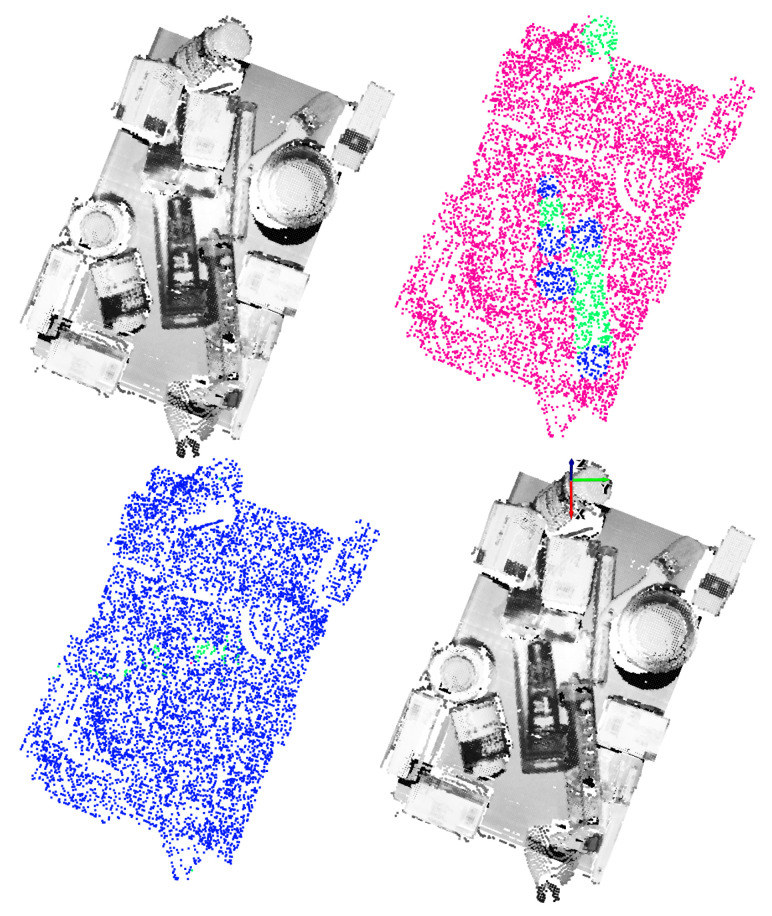
Gripper result example for n=5. Top-left: point cloud of the scene. Top-right: ground-truth annotation. Bottom-left: Top-1% predictions. Bottom-right: Top-1 grasping point. The vertical orientation for the gripper is determined by the *y* (green) axis.

**Figure 16 sensors-21-00816-f016:**
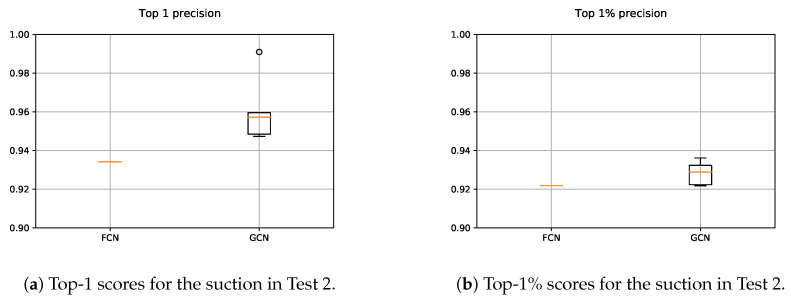
Obtained precision scores per confidence percentiles for the suction models in Test 2.

**Figure 17 sensors-21-00816-f017:**
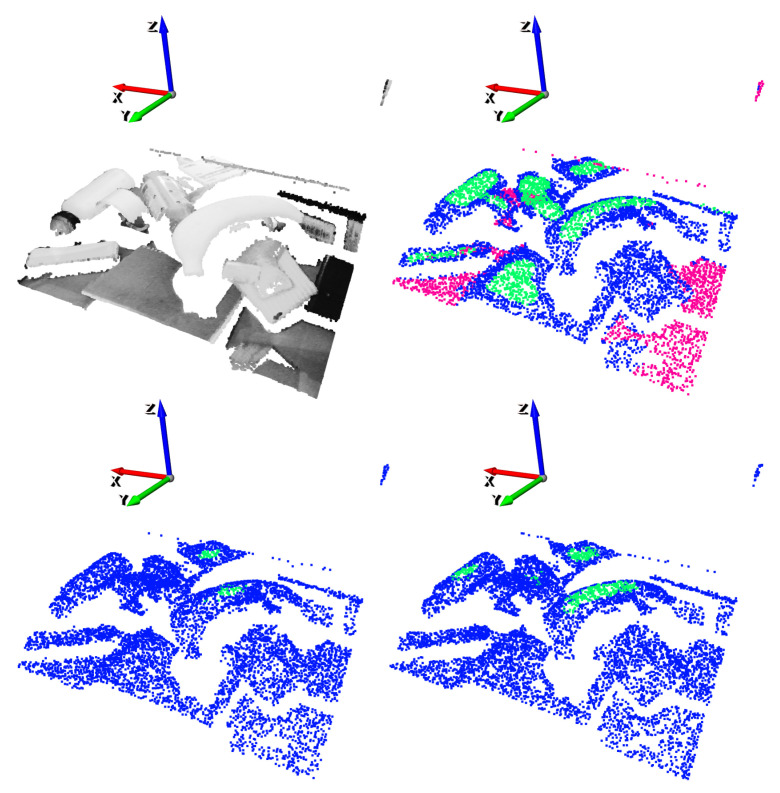
Suction result example with totally new objects. Top-left: point cloud of the scene. Top-right: ground-truth annotation. Bottom-left: top-1% predictions. Bottom-right: top-5% predictions.

**Figure 18 sensors-21-00816-f018:**
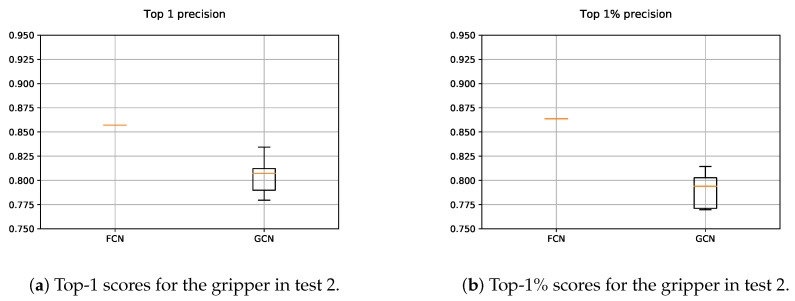
Obtained precision scores per confidence percentiles for the gripper models in Test 2.

**Figure 19 sensors-21-00816-f019:**
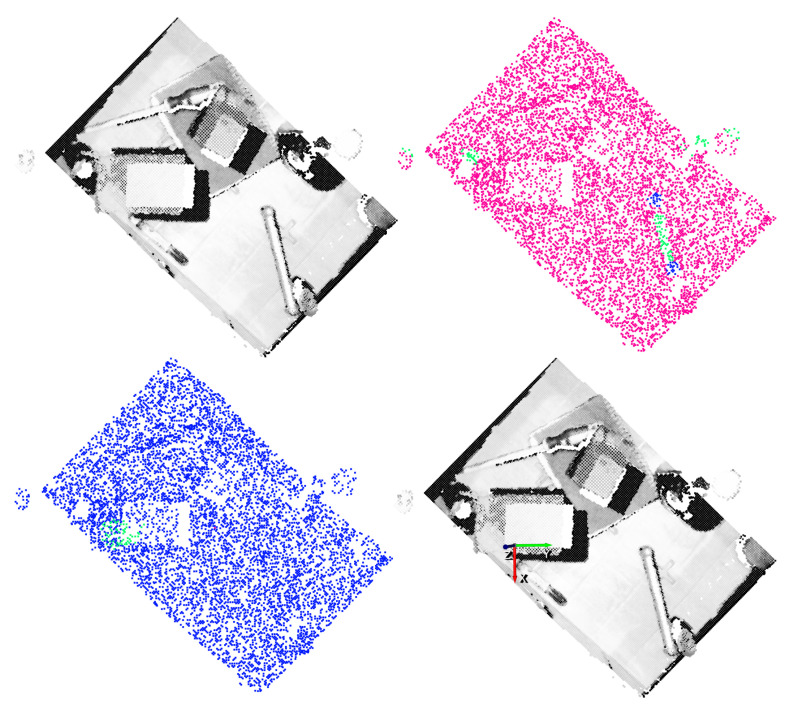
Gripper result example for n=2. Top-left: Point cloud of the scene. Top-right: Ground-truth annotation. Bottom-left: Top-1% predictions. Bottom-right: Top-1 grasping point. The vertical orientation for the gripper is determined by the *y* (green) axis.

**Figure 20 sensors-21-00816-f020:**
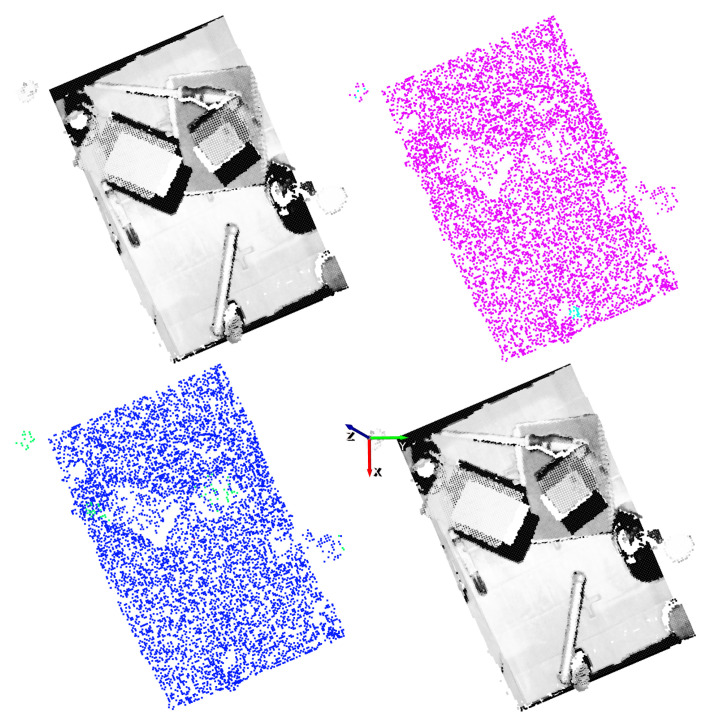
Gripper result example for n=3. Top-left: point cloud of the scene. Top-right: ground-truth annotation. Bottom-left: top-1% predictions. Bottom-right: top-1 grasping point. The vertical orientation for the gripper is determined by the *y* (green) axis.

## Data Availability

The data presented in this study are available on demand from the corresponding author.

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
