# Peer review of "Affordance-Based Grasping Point Detection Using Graph Convolutional Networks for Industrial Bin-Picking Applications"

_sensors, 2021, doi:10.3390/s21030816_

Round 1

Reviewer 1 Report

Dear Authors,

The manuscript is very well structured and organized, has a very good state of the art presentation and the topic is of high interest. However, there are some minor issues to be solved:

- In the abstract, you state that your method significantly improves the performance of such a system. Please state what were the best results, in terms of precision, obtained in the presented literature review and compare them with your results.

- In Figure 5, for Data Pre Processing line, the 3D point cloud representation is for the RGB image in Figure 3, not for the RGB-D image in Figure 5. Please check.

- In line 360, specify the L, l and H dimensions.

- In Figure 6b, as well as Figures 8, 12, 14, 15, it is not very clear which are the axis. Please specify the x,y and z axes in order to have a better understanding.

- In Figure 7, please improve the quality of the picture or make a zoom in order to see which are the good/bad lines.

- In Figure 11, 13, 16, 18 you have the same title and caption for both a) and b) parts of the Figures. Is this correct?

I suggest the acceptance of this paper after solving the minor issues.

Author Response

Dear Authors,

The manuscript is very well structured and organized, has a very good state of the art presentation and the topic is of high interest. However, there are some minor issues to be solved:

Dear reviewer, thank you for the comments on the paper and your suggestions. We have made some modifications to the manuscript following your suggestions (The changes are highlighted with green colour).

- In the abstract, you state that your method significantly improves the performance of such a system. Please state what were the best results, in terms of precision, obtained in the presented literature review and compare them with your results.

We have included the obtained results comparing to the analysed FCN based version.

- In Figure 5, for Data Pre Processing line, the 3D point cloud representation is for the RGB image in Figure 3, not for the RGB-D image in Figure 5. Please check.

Yes, It was a mistake. It is already corrected.

- In line 360, specify the L, l and H dimensions.

We added more specifically the dimensions of the bin.

- In Figure 6b, as well as Figures 8, 12, 14, 15, it is not very clear which are the axis. Please specify the x,y and z axes in order to have a better understanding.

We have modified most of the figures to include the name of each axis. We also think that will help to have a better understanding.

- In Figure 7, please improve the quality of the picture or make a zoom in order to see which are the good/bad lines.

We have zoomed a patch of the image to correctly see the annotations.

- In Figure 11, 13, 16, 18 you have the same title and caption for both a) and b) parts of the Figures. Is this correct?

We have added more specific captions to each of the images.

I suggest the acceptance of this paper after solving the minor issues.

Reviewer 2 Report

The authors present an affordance-based grasping point detection using Graph Convolution Networks (GCN) for industrial bin-picking applications. The literature coverage for GCN is good, and other types of works are also cited well. However, some recent citations good improve the references. The results show good promise. 

However, there are few major concerns.

1) How is the manual annotation done? How can a person separate a good grasp point from a bad one when a physical grasping and pick up is not done using that point using a robot gripper? Simulation-based grasp selection has merit of a physics-based simulation to understand the pickability of the grasp points. On the other hand, manual grasp point selection can appropriately select points based on the type of gripper (two finger vs suction and so on) but cannot effectively differentiate between a good and a bad point. A justification and description of this is required, since the evaluation and judgement of effectiveness of the methods is based on this.

2) Is there an ablation study done to show the effectiveness of the dilated aggregation or the residual connections? It is important to understand the effectiveness of these modifications in the current context.

3) The goal of the paper is confusing. Comparing to FCN, GCN shows improvement, which is understandable. However, there are a number of methods proposed in literature based on PointNet and PointNet++ which show very good performance (both sparse and dense - pixel level) based on point cloud input. There was no comparison to any of them in the literature. Since the proposal is to use (deep) GCN, a comparison to methods based on PointNet architecture is important and relevant.

Author Response

The authors present an affordance-based grasping point detection using Graph Convolution Networks (GCN) for industrial bin-picking applications. The literature coverage for GCN is good, and other types of works are also cited well. However, some recent citations good improve the references. The results show good promise. 

Dear reviewer, thank you for the comments on the paper and your suggestions. We have made some modifications to the manuscript following your suggestions (The changes are highlighted with green colour).

However, there are few major concerns.

1) How is the manual annotation done? How can a person separate a good grasp point from a bad one when a physical grasping and pick up is not done using that point using a robot gripper? Simulation-based grasp selection has merit of a physics-based simulation to understand the pickability of the grasp points. On the other hand, manual grasp point selection can appropriately select points based on the type of gripper (two finger vs suction and so on) but cannot effectively differentiate between a good and a bad point. A justification and description of this is required, since the evaluation and judgement of effectiveness of the methods is based on this.

In the manuscript we state that we have not physically used the gripper but this may be confusing and we have clarified it. Here the point is that we have tested individually how each object is grasped with the real gripper, and used this information to annotate more complex scenarios manually. We decided to do it manually following the idea of the team that won the Amazon Robotics Challenge due to the following reasons:

    • To avoid the usage of 3D models of the parts
    • To take into account the center of mass and weight distribution of the objects that is difficult to do it automatically in simulation (Unless each part is accurately simulated, which is intractable in industrial setups).
    • Although the grasps of the two finger gripper can be simulated, it is difficult to simulate suction grasps (porous surfaces, semi rigid objects...). Manually annotating the dataset we can take it into account.
    • Entanglements between objects are difficult to take into account in simulation unless the physical properties of each object are accurately simulated
    • To avoid the simulation-to-reality gap due to the complexity to create realistic simulations

2) Is there an ablation study done to show the effectiveness of the dilated aggregation or the residual connections? It is important to understand the effectiveness of these modifications in the current context.

Yes, in the original publication of the Deep GCN, they analyse the effectiveness of each of the proposed novelties, including the dilated agregation vs without dilated aggregation and, residual connections vs dense connections vs no skipping layer connections. In the benchmark, the best results are obtained with the model that includes both dilated aggregation and residual connections. We have crarified in the manuscript why did we choose this model configuration.

3) The goal of the paper is confusing. Comparing to FCN, GCN shows improvement, which is understandable. However, there are a number of methods proposed in literature based on PointNet and PointNet++ which show very good performance (both sparse and dense - pixel level) based on point cloud input. There was no comparison to any of them in the literature. Since the proposal is to use (deep) GCN, a comparison to methods based on PointNet architecture is important and relevant.

We decided to compare our method with the FCN based approach because we followed the idea of this work and we used the same encoding for the grasps, annotations … (That is why we have annotated once the dataset and used it to train both FCN and GCN based methods and see whether the results improve using n-dimensional spatial data) and also because it is designed for a bin-picking application. However, as you indicate it would be interesting to make a comparison with other GCN based methods. Since all the reviewed methods were designed only to handle grasping points for the gripper and are not designed for a bin-picking application, it is not straightforward to adapt them to our problematic, and that’s why we have included it as further work.

Reviewer 3 Report

In this paper, the Deep GCN model is adapted that was designed for scene segmentation problems to predict object affordance scores for suction and gripper end effectors. A dataset composed of industrial bin-picking scenarios with randomly arranged multi-reference parts are created to train the GCNs. A Photoneo Phoxi M camera and obtained highly accurate 2D/3D acquisitions of multiple scenes is used.

Author Response

Dear reviewer, thank you for your comments on the paper and the possitive evaluation of it.

Round 2

Reviewer 2 Report

The authors clarified most of the concerns raised. However, it is not entirely convincing that manual labelling is enough for bin-picking clutter where interactions between objects can be easily predicted. On the contrary, a number of new physics-engine based simulators can predict entanglement, occlusion, and object interactions while picking up objects. Of course, without a CAD, simulation-based methods are not usable, which is a separate issue.

For a future work, the authors are suggested to looking into extending a pre-trained GCN (on simulation) to real-data (manual labelled/no label) for new objects, which may be more practical as well.

With the concerns addressed, the manuscript can be accepted in its present form.